# Tissue Engineering Applied to Skeletal Muscle: Strategies and Perspectives

**DOI:** 10.3390/bioengineering9120744

**Published:** 2022-11-30

**Authors:** Ana Luisa Lopes Martins, Luciana Pastena Giorno, Arnaldo Rodrigues Santos

**Affiliations:** 1Centro de Engenharia, Modelagem e Ciências Sociais Aplicadas (CECS), Universidade Federal do ABC, São Bernardo do Campo 09606-070, SP, Brazil; 2Centro de Ciências Naturais e Humanas (CCNH), Universidade Federal do ABC, Bloco Delta, Sala 204, Alameda da Universidade, s/n, Anchieta, São Bernardo do Campo 09606-070, SP, Brazil

**Keywords:** muscle regeneration, bioresorbable polymers, tissue regeneration, spinal muscular atrophy, muscular dystrophy

## Abstract

Muscle tissue is formed by elongated and contractile cells with specific morphofunctional characteristics. Thus, it is divided into three basic types: smooth muscle tissue, cardiac striated muscle tissue and skeletal striated muscle tissue. The striated skeletal muscle tissue presents high plasticity, regeneration and growth capacity due to the presence of satellite cells, quiescent myoblasts that are activated in case of injury to the tissue and originate new muscle fibers when they differentiate. In more severe deficiencies or injuries there is a loss of their regenerative capacity, thus compromising the body’s functionality at different levels. Tissue engineering studies the development of biomaterials capable of stimulating the recovery of cellular activity in injured body tissues, as well as the activity of cells with muscle differentiation potential in injury repair. However, the need for three-dimensional re-assembly in a complex organization makes it difficult to mimic this tissue and fully regenerate it for the sake of precise and effective movements. Thus, this article aims to provide a narrative review of tissue engineering strategies applied to the regeneration of skeletal muscle, in a critical evaluation of research, whether aimed at injury or atrophies such as spinal muscular atrophy.

## 1. Introduction

The entire healthy human body is composed of a complex system that gives the individual functionality and enables movement. From the Latin movere, the word “movement” represents a form of expression and the change from one state to another.

The muscle tissue is an integral part of this synergy and it is derived from the mesenchyme, embryonic connective tissue, which originates from the mesoderm, except in the muscles of the iris, which originates from the neuroectoderm. After the myoblasts differentiation (embryonic muscle cells) there is the formation of elongated and contractile cells with specific morphofunctional characteristics; they are classified as smooth, cardiac striated, and skeletal striated muscle tissue [1].

Highlighting the striated skeletal muscle tissue as the most abundant in the body, it identifies an organizational structure composed of epimysium, perimysium, and endomysium, through which blood capillaries, lymphatic vessels, and nerve endings flow. Each muscle fiber (or myocyte) is made up of elongated, multinucleated, cylindrical cells joined by connective tissue sheaths that are organized into a repetitive arrangement of sarcomeres, actin, and myosin filaments (myofibrils) [1,2].

Although this tissue has post-mitotic cells, in other words, cells that do not divide, there is the presence of satellite cells that assist in tissue repair. These quiescent cells are located adjacent to the muscle fiber, in the adult and may return to self-renewal and proliferation capacity in response to stress or injury [1,3].

The molecular markers of the satellite cells are the transcription factors Pax-3 and Pax-7 (paired box protein-3 and 7, respectively). Once the normal tissue architecture is disrupted, the satellite cells are exposed to regenerative signals that stimulate them out of their resting state [3]. Satellite cell myogenesis in response to injury begins with the activation of local signals that positively regulate myogenic differentiation transcription factor 1 (MyoD) and myogenic regulatory factor 5 (Myf5). As so, a myoblastic lineage is determined. The myoblasts enter the proliferative phase, differentiate into myocytes, and fuse into multinucleated myotubes after activation of the transcription factor myogenin. After this terminal differentiation, the myotubes fuse into myofibers [3,4,5]. However, satellite cells have types with distinct characteristics from each other, so cells present in different anatomical locations in muscle tissue may have effective differentiation potentials. Thus, about 80% of the satellite cells constitute a cell group called responsive cells, which have a faster and more effective differentiation capacity while the other 20% of the cell population is composed of the reserve cells which exhibit slow differentiation in response to tissue injuries. This is probably due to the microenvironment with specific molecular characteristics where these cells are embedded [6,7].

Satellite cells confer limited capacity for tissue repair. External events or diseases of the most diverse etiologies, depending on their magnitude, can cause muscle injuries that exceed the endogenous capacity of regeneration compromising the muscle tissue [8].

Injuries related to sports, domestic accidents, or even automobile accidents of moderate to severe implication usually require surgical procedures, such as the implantation of muscle flaps and composite tissue grafts. In the first method, muscle fragments are taken from autologous muscle tissues along with their nerve and blood supply, to be implanted at the injured site. This procedure is very limited by the amount of tissue that can be removed from other parts of the body without causing compromise at the donor site, rarely results in complete functional recovery, and occasionally results in fibrosis rather than myogenesis [5,9,10]. In the second method, allograft is used, such as transplanting skeletal muscle, skin, bone, and tendon from a donor that is genetically different from the patient. In this procedure, one of the limiting factors refers to compatibility, which can lead to an exaggerated immune response from the receptor. In this case for some successful therapy the patient will need continuous immunosuppression [5].

On the other hand, abnormalities in the organization of the muscle fibers present themselves as myopathies leading to atrophy or degeneration of the muscle fiber. Muscular dystrophies (Duchenne and Becker, for instance) and spinal muscular atrophy (SMA) are neuromuscular diseases of genetic origin. In the first case we have a mutation in genes that affects muscle tissue directly (as in dystrophin in the case of Duchenne muscular dystrophy), which leads to atrophy and death of muscle cells. In the second case, alterations in motor neuron genes in the spinal cord are observed, which can lead to loss of function, loss of self-image, and implications for self-esteem [10,11].

As new pathological mechanisms are being discovered, several therapeutic strategies are also being explored, most notably tissue engineering [12,13]. The complex organization of muscle tissue with its various cell types mobilizes the customization of therapy to improve the patients’ quality of life, using natural or synthetic materials as plausible inputs in the desired stimulation of full tissue regeneration [14].

In this regard, tissue engineering aims to construct scaffolds associating cells with different potential differentiation that attempt to somehow mimic the tissue environment and the factors which stimulate the cells in their proliferation and differentiation. In short, with this approach and with a level of interdisciplinarity, one seeks to recreate functional organic tissues without causing side effects when interacting with the organism [15].

Complex or extensive injuries in skeletal muscle, which exceed the normal capacity of regeneration, in theory, could be repaired through devices associating cells, biomaterials, and growth factors. Once inserted into injured individuals, these tissue-engineered devices can interact with the body and take over the function of the impaired cells to restore the locomotor system’s activity [10,12]. Experimental results have shown promising results using scaffolds manufactured from extracellular matrix (ECM) components, hydrogels, and 3D bioprinting products [12]. Theoretically, the same approach could also be considered for the most benign forms of dystrophies or even SMA.

The three-dimensional environment provided by biomaterials is important to offer conditions for the differentiation of myogenic cells. Trying to mimic this seems important to us. Skeletal muscle is a dynamic tissue responding to mechanical stimuli by atrophy with decreasing mechanical stimulation or conversely with hypertrophy following the application of increased mechanical stresses [1,2,3,4,8]. Comparatively, mechanical stimuli play an important role in myogenesis tissue engineering. The same is true for muscle atrophy that occurs without electrical/nerve stimulation. Electrical stimuli are important for the muscle cell. All these factors must be taken into account in order to build a three-dimensional environment that stimulates muscle differentiation.

This narrative review aims to present tissue engineering procedures applied to skeletal muscle regeneration, regardless of the type of injury or atrophy, such as SMA. We analyzed myogenic cells and their characteristics, the biomaterials used for muscle tissue engineering (basically natural or synthetic polymeric materials), and finally the three-dimensional environment for myogenesis. We present the most important results and their limitations.

## 2. Materials and Methods

The literature review was carried out using the PubMed, LILACS, Scopus, Web of Science, and SciELO databases. The Health Sciences Descriptors (DeCS) were Muscle Regeneration, Bioresorbable Polymers, Tissue Regeneration, Spinal Muscular Atrophy, Muscular Dystrophies. Boolean operators (AND, OR and NOT) were also used. After reviewing the articles, we evaluated publications from the last twenty-one years selecting the key concepts of each researched work, and grouping them by similarities in the information provided by the established authors. Afterwards, we developed the final text.

## 3. State of Art

Biomaterials developed with characteristics that mimic organic tissues and with the ability to promote an environment conducive to angiogenesis and myogenesis can be used to regenerate injured muscle tissues. In addition to creating a favorable environment for tissue regeneration and an ideal for the development of new cells, the three-dimensional biopolymers must be biologically compatible to minimize the body’s immune reactions [16].

Kwee and Mooney [16] report two approaches to muscle tissue engineering: (a) the use of biomaterials developed with muscle and supporting cells, cultured in vitro, in order to mold the injured tissue fully and then be implanted in the injury site and; (b) the so-called in situ approach, in which chemical signals, enzymes, and growth factors are combined in 3D matrices aiming to activate the organism’ own satellite cells and organize the endogenous tissue regeneration process after being implanted in the injured site. Moreover, there are also reports of implants such as functionally engineered scaffolds applied directly into the body. These approaches can be seen in Figure 1.

Among the in vitro cell culture methods, the use of myoblasts in a fibrin gel matrix with tensioned fibers demonstrated growth of aligned muscle cells that responded to mechanical stimuli. When using endothelial cells in this system, cultivated in matrixes with fibrin, vascularization was also observed aligned to the fibers of the scaffold. In situ techniques have shown promise in controlling the internal processes of injured tissue regeneration by growth factors and enzymatic signaling, capable of activating cellular processes of tissue repair [16].

**Figure 1 bioengineering-09-00744-f001:**
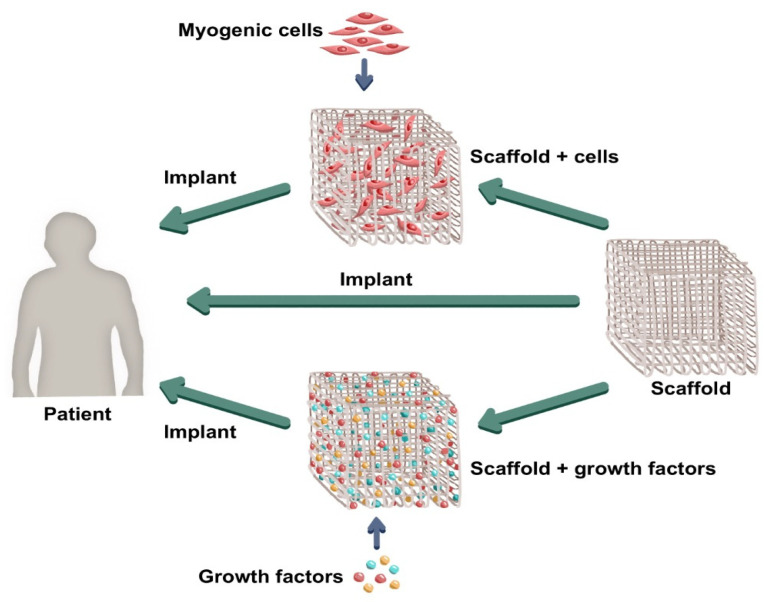
General approaches in muscle tissue engineering. (Source: composed by the authors).

The wide use of the vascular endothelial growth factor (VEGF) (as an angiogenic factor) and insulin-like growth factor 1 (IGF-1) (as a myogenic factor) were observed in the literature, showing the ability to increase myogenesis in injured muscle tissue while maintaining tissue viability and tissue function. It was also observed a recovery of muscle mechanical function subsequent to various types of injuries [17]. Furthermore, porous alginate cryogels with the aminoacyl sequence Arg-Gly-Asp (RGD) and enriched with the same growth factors (VEGF and IGF-1) have been shown to promote in vitro mesenchymal stromal cells to secrete bioactive factors, such as VEGF, MMPs, GM-CSF, IGF-1, HGF, TGF-β, and bFGF, which deeply stimulate the function of muscle progenitor cells. In vivo, this paracrine signaling by mesenchymal stem cells (MSCs) associated with porous alginate scaffold has significantly improved muscle function, with greater recovery of the contraction force and increase in the density of myofibrils, remodeling scar tissue and promoting the formation of new myofibers [18].

Madden and colleagues [19] have reported the production of in vitro skeletal muscle fibers capable of mimicking the shape and function of the original tissue. Primary myogenic cells cultured in a gelatinous scaffold were used as a study model and once differentiated they formed aligned multinucleated cellular fibers with functionally good contraction results in response to different chemical and electrical stimuli. The myotic bundles were developed from the human own muscle satellite cells in a hydrogel mold (fibrin/matrigel) mimicking the ECM. The use of these cells in an environment conducive to cell differentiation gave rise to a skeletal muscle tissue in a dense three-dimensional environment containing fibroblasts, new satellite cells with differentiation potential and biological signaling pathways necessary for muscle contraction [19].

Calcium flow monitoring tests were performed with the aim of evaluating the capacity and potency of fiber contraction developed in response to chemical and electrical stimuli, so that greater intensity and speed of calcium flow in the cells signaled a more vigorous muscle contraction in response to the given stimuli. The use of higher intensity electrical stimuli promoted stronger contraction than stimuli of lesser intensity. Additionally, some types of drugs were tested mainly to indicate the ability of each one to promote contraction as well as the possible effects and chemical damage they can cause in in vitro muscle model. The results suggested that the use of statins (simvastatin and lovastatin) causes a long-term decrease in muscle contractile function and promotes lipid accumulation in the tissue. The use of chloroquine showed similar results regarding the reduction of muscle contraction potential. On the other hand, clembuterol, when used in low concentration, showed an improvement in muscle contraction and hypertrophic activities, with no harm to the tissue [19].

However, in vitro-grown tissues may not always guarantee the organization into fibers and the recovery of the typical functions of muscle tissue. Further research is still needed to discuss the compatibility of the developed tissues with the organism in which they will be implanted, as well as how they will perform when they replace the original tissues. Nevertheless, the findings by Madden et al. [19] have shown that there is significant potential for in vitro tissue development using the myogenic cells present in human muscle tissue. This same system has been shown to be responsive to drugs, indicating an in vitro model for the study of drugs targeting muscle tissue [19]. This approach can also be interesting in trying to recover muscle damage caused in individuals with less severe muscular dystrophies and even in individuals with atrophies, such as SMA.

### 3.1. Cells with Myogenic Potential

Cells with myogenic potential are the basis of tissue engineering applied to skeletal muscle. These topics are discussed in the following text.

#### 3.1.1. Satellite Cells

Satellite cells from muscle tissue could differentiate and originate new myocytes, and are shown to be the ideal cell type to be used in the development of in vitro muscle tissue. Due to the difficulty of isolating the cell type coming from human muscle tissue itself, research conducted by Syverud et al. [20] studied efficient ways to obtain and culture satellite cells to improve the development of biomaterials to replace the muscles of individuals with severe tissue injuries.

Among the methods used to obtain satellite cells from biological donors, it is difficult to guarantee both the required number of cells and the purity of the samples. The isolation methods of these myogenic cells can be conducted by: (a) individual explant, a method that ensures great purity of each cell, but makes it difficult to obtain the amount needed for the development of biomaterials and; (b) enzymatic dissociation, a method that promotes the acquisition of a greater amount of satellite cells but with other cellular elements from the action of the enzymes applied, requiring the later use of purification techniques of the samples obtained [20].

Techniques to isolate satellite cells from the sample can be performed on culture plates, where the samples are left until the satellite cells adhere to the plate and the other cell types remain in the suspension so that they can then be discarded. A more efficient purification technique can be used with Fluorescence Activated Cell Sorting (FACS) and Magnetic-Activated Cell Sorting (MACS). FACS operates by laser identification of cell markers conjugated with fluorophores and applied to physically separate a specific cell type [21]. On the other hand, MACS works by incubating cells with antibodies to specific markers conjugated to magnetic microbeads [22]. These two techniques are used to distinguish myogenic cells from other cell types found in the samples. However, until now, the absence of a specific and definitive surface marker, these methods may not adequately differentiate the cells of interest and promote less effective sample purification [20].

After collecting satellite cell samples from muscle tissue, several types of growth factors and biological signals are required to develop an environment like that found in vivo, to cultivate them in an environment conducive to proliferation and differentiation in vitro. Several types of hydrogels or cell matrices can be used as scaffolds that promote both the support of the biopolymer and the flow of substances, ensuring the resumption of the shape and function of the cultured tissues when implanted in the body [20].

Despite showing great potential in originating muscle fibers in vitro due to their characteristics and biological function, satellite cells have important practical limitations. These cells are obtained through skeletal muscle biopsy, which, besides being a painful and invasive process, only a small portion is clinically acceptable. Both the obtention and purification processes are laborious: the obtention by enzymatic dissociation is costly and hardly allows, simultaneously, the collection of samples with high purity and a significant number of cells, and the purification requires complex methods such as FACS and MACS. In addition, satellite cells have limited differentiation potential, their ability to duplicate outside biological tissues is limited and in vitro forms of preservation are still poorly studied. Considering these difficulties in using myogenic muscle tissue cell management, other cell types can be used as alternative cellular raw material for laboratory-developed scaffolds [5].

#### 3.1.2. Embryonic Stem Cells (ESCs)

ESCs are derived from the inner cell mass and have the potential to differentiate into cell types from the three primordial layers of the embryo (endoderm, mesoderm, and ectoderm) [23]. Thus, if properly stimulated, they have the capacity to originate all cell lines, proving to be ideal for the development of tissues in vitro. However, research on the potential use of ESCs in tissue engineering has run into ethical issues due to the means of obtaining this cell population, since they are isolated from human embryos. Furthermore, ESCs have the possibility of forming tumors after being implanted in the body. Because of the difficulties involving the use of embryonic stem cells, other stem cell populations are gaining prominence in studies about regenerative medicine [24].

In Brazil, it is worth mentioning that Law 11.105/2005 [25] establishes safety norms and inspection mechanisms for activities involving genetically modified organisms [...], creates the National Biosafety Council [...], restructures the National Biosafety Technical Commission and provides on the National Biosafety Policy. Additionally, the Collegiate Directives Resolution (RDC) number 505 [26], 506 [27] and 508 [28] of 2021 that addresses the registration of advanced therapy product, cites rules for conducting clinical trials with investigational advanced therapy products and addresses good practices in human cells for therapeutic use and clinical research, respectively, indicating discussions in this direction and moving towards the fourth generation of biomaterials.

#### 3.1.3. Multipotent Stem Cells

Multipotent stem cells obtained from human amniotic fluid (amniotic fluid stem cells, or AFSCs) also show potential for use as precursors to muscle cells, as they possess the ability to differentiate into all somatic cell lineages and have high proliferation potential. In fact, AFSCs appear to have properties intermediate between ESCs and adult stem cells [29]. When AFSCs are implanted in the body, they show low immunological rejection and less potential for tumor formation when compared to ESCs, moreover, there are no ethical issues surrounding their procurement process. Although there are few studies on the use of this cell type in tissue development in the laboratory, AFSCs are cells with promising capabilities in the tissue engineering field [24,29].

In addition to the use of pluri and multipotent cells obtained in stages prior to cell differentiation into specific lineages, an alternative would be the use of already differentiated human tissue cells genetically reprogrammed to return to the initial stages of differentiation to make them undifferentiated stem cells again; these are called induced pluripotent stem cells (iPSCs) [30]. This cell reprogramming technology, for generating iPSCs, presents a promising method, however, the difficulty of producing a significant amount of stem cells with an adequate degree of purification make it difficult to use this method on a large scale [24]. iPSCs are very similar to ESCs in terms of differentiation capacity, however, iPSCs do not require the destruction of embryos for their production. As well as ESCs, the main safety issue regarding iPSC-based therapy is the risk of teratoma formation. It is also important to highlight that due to the genomic instability of iPSCs, even the improved protocols for their differentiation do not guarantee a safe clinical application [31]. See Figure 2.

##### Mesenchymal Stem Cells (MSCs)

MSCs are multipotent cells that exhibit a high degree of plasticity and can be used as raw material for muscle tissue development in vitro. Possibly originating from a more undifferentiated population of fibroblasts and pericytes, MSCs are present in red bone marrow, but cell types such as MSCs can be found in other sources, such as dental pulp, adipose tissue, placenta, and umbilical cord [32]. However, studies using MSCs from different tissues have shown that they comprise a heterogeneous group of cell populations, each with its own differentiation potential. The ability of each population to give rise to distinct cell groups in the organism is related to the in vivo environment from that each one originates, so that they have their own molecular markers that are expressed in the form of specific marker proteins. Although the subpopulations of MSCs have distinct differentiation potentials among each other, they all possess the ability, to a greater or lesser extent, to give rise to all cell types if exposed to specific conditions when cultured in vitro [32]. The MSCs strain for muscle differentiation can be seen in Figure 2.

Classically, MSCs are defined based on three criteria: (a) remain adherent to plastic under cell culture conditions; (b) express the CD105, CD73, and CD90 receptors and do not express the markers CD45, CD34, CD14 or CD11b, CD79a or CD19 and HLA-DR and (c) differentiate into osteoblasts, adipocytes, and chondrocytes in vitro [33,34]. Lv et al. [32] critically studied many other markers present on the surfaces of MSCs populations in an attempt to find both specific markers that characterize a subpopulation and a common marker across all their lineages. The goal was to characterize MSCs and distinguish them from other cell types present in the body. Of the many markers studied, highlighting Stro-1, CD271 SSEA-4, CD-146, CD49f, CD349, GD2, 3G5, and SUSD2, many have been shown to be relevant and present in many subpopulations of MSCs, where Stro-1 and CD 146 have been shown to be quite comprehensive. The other receptors mentioned were also shown to be present in many subpopulations, even allowing separation of some of them. Nevertheless, a universal marker for MSCs has not been found [32]. The range of receptors in the different cell types with myogenic potential can be seen in Table 1.

The transmembrane glycoprotein CD34 is an important point. In a report review, Sidney et al. [35] indicated that CD34 may also be considered as an important marker in some stem cell types of various tissues, especially in progenitor cells. First identified in hematopoietic stem cells (HSCs), the function of CD34 was initially associated with adhesion, regulation of cell differentiation and proliferation, as well as some role in the migration of hematopoietic cells into specific niches within the bone marrow. However, the glycoprotein is found in several cell types of non-hematopoietic tissues in MSCs from adipose tissue (although not from bone marrow MSCs), in muscle satellite cells, stromal cells, and in epithelial tissue progenitor cells, which have differentiation potential for tissue regeneration. Although all these cell types express CD34, not all exhibit the same properties. Many co-express specific markers alongside CD34, suggesting that the presence of CD34 could indicate a progenitor stage for that tissue [35]. Because CD34 is present in the satellite cells of skeletal striated muscle, it may be important in the context of muscle tissue engineering. The different properties observed in muscle satellite cells can be explained by the presence of distinct subsets of CD34 cells, with distinct differentiation potentials. For example, CD34+ cells that co-express the endothelial marker CD31+ show angiogenic differentiation. However, CD34+ and CD31- cell populations show greater potential to differentiate into adipogenic and myogenic lineages [36]. Thus, although the function of the CD34 antigen is not yet elucidated, the fact that it is expressed in stem cell populations of various tissues demonstrates that its function is probably associated with the processes of regulation, differentiation, and cell proliferation [35,36,37,38,39,40]. The most commonly used MSCs populations are those derived from adipose tissue and bone marrow.

**Table 1 bioengineering-09-00744-t001:** Molecular markers in different myogenic stem cell populations compared to hematopoietic stem cells.

Markers	Cell Types				References
	Satellite Cells	BM-MSCs	ADSCs	HSC	
CD73		+			[33,37,38]
CD90	+	+	+	+	[34,35,37,38,40]
CD 105		+			[33,37,40]
Stro-1		+	+		[32,33,35,38]
CD 73		+			[33,38]
CD 146		+			[32,38,40]
SSEA-4		+			[32]
GD 2		+	+		[32]
CD 49D			+		[34]
CD 49F		+			[32]
CD34	+		+	+	[35,37,38,39,40]
CD45				+	[35,37,38,40]
HLA-DR				+	[37]
CD 38				+	[35]
CD 117				+	[35]
CD 133				+	[35]
CD 56	+				[35]
Myf5	+				[35]
Desmin	+				[35]
M-Cadherin	+				[35]
CD106	+	+			[34,35,38]
Flk-1	+				[35]
VEGFR	+				[35]
MyoD	+				[35]
CD146	+				[35]
CD73		+			[33,37,38]

BM-MSCs = bone marrow mesenchymal stem cells; ADSCs = adipose tissue-derived stem cells; HSC = hematopoietic stem cells; + is the expression of the marker.

##### Adipose Derived Stem Cells (ADSCs)

CD34 has been identified in sample-obtained ADSCs, but the number of markers tends to decrease rapidly after cell extraction, most likely due to the cellular response to a change in medium. ADSCs have the capacity to differentiate into other cell types of mesodermal lineage, and can originate new muscle cells, being an alternative form of raw material to produce muscle tissues [41]. In addition to being abundant in the body, ADSCs express little of the major histocompatibility complex class I (MHC I) and do not express major histocompatibility complex class II (MHC II), so both allogeneic and xenogeneic samples can be used in in vitro tissue development without causing a marked immune response in the individuals in whom the tissues will be implanted, thus avoiding the need for immunosuppression [5,42]. In general, adult human MSCs from different sources express intermediate levels of MHC I, but do not express class II human leukocyte antigens (HLA) on the cell surface. The expression of HLA class I in human fetal MSCs is even lower [43]. The immunological phenotype of MSCs with their expression/non-expression characteristics of markers (MHC I+, MHC II−, CD40−, CD80−, CD86−) is considered non-immunogenic and therefore transplantation into a halogenic host does not require immunosuppression. MHC class I could activate T cells, but with the absence of co-stimulatory molecules, a secondary signal would not be activated, leaving the host T cells without activity [44].

The techniques for obtaining, isolating, purifying and growing ADSCs offer advantages when compared to those of satellite cells. Through liposuction, ADSCs are easily obtained cells and their samples can be collected in large quantities [45]. Isolation methods are efficient, through enzymatic digestion, using collagenase type A and fetal bovine serum. Purification of the samples can be conducted by the already mentioned FACS and MACS processes, however, it is less efficient due to the lack of universal markers that differentiate adipose tissue-derived stem cells from other cell types. The use of ADSCs can be seen in Figure 2.

Currently, research shows that explant culture techniques can also be used to isolate cells and obtain purer samples of adipose tissue-derived stem cells, eliminating the need for further purification. After sample collection and purification, cryopreservation has been shown to be the most suitable method of preserving ADSCs to maintain their myogenic potential [5].

Among the ways to induce the differentiation of ADSCs in myoblasts are biochemical, physical, and genetic stimuli. Non-myogenic cells of different origins, including muscle-associated or bone marrow-derived cell populations, have been shown to be able to form skeletal muscle fibers when co-cultured with myoblasts. The same seems to happen with the adipose tissue-derived population, since the simple direct contact between ADSCs and myoblasts can induce their fusion into myotubes, promoting cell differentiation and contributing to skeletal muscle regeneration [11]. This suggests that cell-cell communication can stimulate the myogenic differentiation of ADSCs. Furthermore, it has been shown that ADSCs have a high regenerative capacity in vivo as they can be incorporated into muscle fibers after ischemia and can significantly restore dystrophin expression in mdx mice, an animal model for Duchenne muscular dystrophy [11]. The periodic application of mechanical tensioning on human ADSCs has demonstrated the induction of cell alignment, cell fusion, formation of myotubules with an increase in the number of cell nuclei and the expression of myogenic proteins, contributing to cell differentiation [46]. Changes in gene expression using viral vectors tested in mice to induce MyoD protein production were successful in increasing myogenic induction of cells and direct application models of MyoD in cattle showed similar results and increased viability in in vitro tissue development [5].

Studies comparing the results obtained with tissue implants developed from satellite cells (or muscle progenitors, for some authors) of skeletal muscles and from ADSCs revealed similar potentials for muscle regeneration and formation of new tissues and the result obtained with the use of ADSCs presented advantages related to better vascularization [47]. An important point in this study is that decellularized extracellular matrices (dECM, which will be discussed in item 3.2.3) were used as a vehicle for cells, so that arrays with populations of muscle progenitor cells and ADSCs were inserted into mice. The scaffolds with muscle cells exhibited a more differentiated phenotype (immunohistochemical markers used: myosin, titin, junctophilins and ryanodine receptor 1), with myotubes interspersed with some unfused myogenic cells, while the matrices with ADSCs exhibited mainly an undifferentiated phenotype. Data suggest that differentiation of ADSCs occurs best in the presence of co-cultured muscle cells. Furthermore, it was taken full advantage of the paracrine effect of undifferentiated ADSCs [47]. This proved effective in the outcome, leading to better vascularization.

##### Bone Marrow Mesenchymal Stem Cells (BM-MSCs)

Mesenchymal stem cells obtained from bone marrow, similarly to other MSCs, have the potential to differentiate into muscle cells, and therefore, can also be used in the development of muscle tissues in vitro (see Figure 2). Despite the limited potential for muscle differentiation of this population of MSCs, when cultured with myoblasts, they fuse to them and can contribute to muscle tissue regeneration [48,49]. Bone marrow-derived MSCs secrete growth factors that contribute to the regeneration, proliferation, and development of myoblasts, such as VEGF, bFGF, IGF-1, and HGF to promote the formation of new muscle cells in a conducive environment when cultured together in 3D scaffolds. Among the growth factors, HGF, and IGF-1 stand out, mainly inducing the proliferation of satellite cells, and activating the proliferation and differentiation of these cells, respectively [49].

Witt et al. [49] examined the effects of IGF-1 and HGF in cell cultures with BM-MSCs and in myoblasts grown alone or in co-culture. HGF in moderate concentration contributed to the proliferation and differentiation of MSCs into muscle cells, but both very low and very high concentrations of both growth factors decrease the development potential of stem cells. This suggests that there is an optimal concentration of these factors to activate muscle differentiation. Samples comparing cultures of isolated myoblasts, isolated MSCs, and of the two associated cell types under the exclusive action of IGF-1 showed that monocultures of myoblasts developed the fastest, possibly due to the length of time MSCs require to initiate their differentiation process [49].

Similar results were observed when testing IGF-1 and HGF, both together and alone, demonstrating good differentiation capacity in the samples. However, the successful differentiation of MSCs into muscle cells is strongly related to their interaction with myoblasts since the fusion of cell types can induce muscle tissue development [49]. BM-MSCs and myoblasts are known for secreting several growth factors involved in the process of muscle regeneration [48]. Then, autocrine, and paracrine stimulation can lead to myogenic differentiation. However, data from Witt and colleagues [49] supports that muscle differentiation depends on cell-cell contact rather than secreted factors and demonstrates that type I collagen gel matrices and polycaprolactone (PCL) were promising for muscle tissue development in vitro. These collagen gel matrices, indicated by the authors [49], are other examples of scaffold types employed in tissue engineering. Moreover, the three-dimensionality of certain scaffolds enables myogenic differentiation maintained by structures composed of polymeric materials, natural or synthetic, or even provided by structures derived from biological tissues themselves, seem to be fundamental for muscle differentiation. Thus, the scaffold used in in vitro tissue production should mimic the endogenous environment in which the cultured cells should be implanted, as well as should allow the cell culture to develop three-dimensionally and in an aligned manner.

A point that we explored earlier and will draw attention to is the work of Di Rocco et al. [11], which brings an animal model for Duchenne muscular dystrophy. There are also reports in the literature of in vitro models for Duchenne [50] or some forms of atrophies [51,52]. So, we understand that we are close to the approaches designed under the dynamics of tissue engineering begin to be proposed for the study and treatment of rare diseases in skeletal muscle.

However, it should be noted that obtaining myogenic cells is just part of the problem. The induction of differentiation is another critical part. Biological, mechanical, and electrical stimuli are used to induce myogenic differentiation (this will be addressed next).

### 3.2. The Three-Dimensional Environment for Myogenesis

As previously cited, in vitro tissues produced from myogenic cell culture can be developed in scaffolds of various materials to ensure adequate structural and biochemical support for the processes of cell differentiation and proliferation [24].

Synthetic polymer scaffolds are attractive due to their versatility, as they can be easily tailored to meet specific mechanical properties, porosity, and degradation time according to the characteristics of the tissue and the lesion site that must be regenerated. Biodegradable polymers such as polyesters, polyurethane and polyamides have been gaining prominence in studies in the area because they have physical and mechanical properties similar to those found in the muscle environment, with high potential for use in the development of muscle tissues in vitro [16,24].

Although synthetic polymers have the advantage of being produced according to planned characteristics and at relatively low cost, they can trigger inflammatory reaction in the organism in which they are implanted due to the possible toxicity of their components. In addition, synthetic polymers lack functional motifs recognized by cells, which makes initial cell adhesion difficult and slower. Therefore, the use of natural biodegradable polymers has advantages due to its biocompatibility towards organism, avoiding relevant inflammatory processes, as well as implementing a three-dimensional environment favorable to cell differentiation and proliferation [24,53].

#### 3.2.1. Methods for Obtaining a 3D Environment

The scaffolds used in tissue engineering can be manufactured by different techniques and have several designs. The matrices must be made up of porous or fibrous structures, preferably with interconnected pores, designed to allow nutrient exchange and blood supply. Hence, the scaffold allows cells to grow in a three-dimensional environment to maximize differentiation and development of the cell type to be cultured in it, as well as to facilitate invasion of the tissue formation [24].

Some techniques used for the fabrication of the scaffolds are electrospinning, rotary jet-spinning, and 3D bioprinting. In electrospinning, an electric field is used to produce the nanofibers that will give rise to the scaffold from the solution of the polymer used as raw material. Kang et al. [54] associated this technique with 3D printing and were able to formulated biopolymers with the potential to create in vitro, tissues with characteristics similar to those of skeletal striated muscle tissue and promote the development of new muscle cells. The fibers of this study were produced with different polymers, such as PCL, gelatin, poly (methyl methacrylate) (PMMA), poly (lactic acid-co-glycolic acid) (PLGA), alginate, fibrinogen, and the association of some of these with each other. By using the electrospinning technique, the study has shown the possibility of synthesizing nanofibers with structural properties similar to the muscle ECM, where one can observe the proliferation and growth of myoblasts often with a well-developed F-actin cytoskeleton, in addition to other muscle cell markers, such as myogenin and MyoD proteins. The cellular study indicated that without biochemical cues, the mechanical and morphological characteristics of electrospun fibers are a crucial factor for determining cellular viability and functionality [54].

In rotary jet-spinning, the fibers are developed by applying a centrifugal force to the base polymer solution. Considering large-scale production, rotary jet spinning tends to be more advantageous than electrospinning, since it allows the optimized production of the fibers used in the formation of scaffolds [55]. Furthermore, Badrossamay et al. [56] highlighted highly anisotropic mechanical properties of the super-aligned nanofibers produced by the respective technique, as well as demonstrated aligned cell morphology, improved cell proliferation and viability, thus indicating another possible advantage of rotary jet-spinning compared to electrospinning in specific applications in biological tissues. Promising results for skeletal muscle tissue engineering using rotary jet spinning were also described by MacQueen et al. [57] as we will see below.

Three-dimensional bioprinting is a more modern biomanufacturing technique that can also be used in the development of scaffolds. Additive manufacturing, commonly known as 3D printing, occurs by a process of manufacturing three-dimensional objects from the controlled deposition of successive layers of a given material in order to form a structure [58]. On the other hand, bioprinting aims at the integration of living cells with biomaterials forming three-dimensional structures, which allows the automated and reproducible production of functional tissues in a three-dimensional way by depositing biocompatible materials layer-by-layer with a high-precision cell positioning. Bioprinting can recreate more complex 3D organ level structures and incorporate mechanical, as well as biochemical cues that are crucial elements of the whole organ architecture [59,60]. However, it is difficult to simulate the mechanical environment present in fabrics today.

Kim et al. [61] showed that matrices made through 3D bioprinting from fibrinogen, hyaluronic acid, and gelatin were able to give rise to an organized structure compatible with endogenous muscle and that favored the maturation and differentiation of the human muscle progenitor cell (hMPC). The bioprinted skeletal muscle tissue was organized in the form of highly organized multilayered bundles of muscle fibers, formed by densely clustered and aligned viable myofiber-like structures. A bioprinted structure allows longitudinal alignment of actin-rich cytoskeleton cells in response to mechanical signals and to maintain the structural integrity of multi-layered muscle. The scaffold produced by this technique allowed a functional recovery of up to 82% of the injured tissue when implanted in rats, demonstrating in vitro tissue innervation and vascularization [61].

In addition to the use of 3D bioprinting for the development of cell culture scaffolds, Bour et al. [62] demonstrated that this technique can also be used to position endothelial cells and pericytes along with myoblasts in the bladder dECM to “pre-vascularize” the scaffold developed for muscle tissue engineering and repair. The results were encouraging, as endothelial cells seeded on the dECM begin to form network-like structures after only 24 h, with high viability (>90%). However, the mechanical behavior of myoblasts was not evaluated in this study, only that they had a desmin-rich cytoskeleton.

The addition of growth factors in matrices made from collagen showed promising results, since it was able to promote myoblast cell differentiation [24]. Among the natural polymer scaffolds, the ones produced using collagen types I, II and III stand out, for presenting good compatibility with the organism and allowing the organization of a three-dimensional fibrous matrix that mimics the one found in the tissue environment [63]. According to Nuge et al. [24], the most suitable technique for the manufacture of this type of matrix would be the electrospinning.

#### 3.2.2. Natural Polymers

The use of a collagen product, gelatin, also presents promising results in the manufacture of scaffolds. Gelatin, an easily obtained raw material, interacts favorably with the cell structure and presents applications in the medical, pharmaceutical and cosmetic areas [56]. However, the production of a gelatinous matrix through the electrospinning technique is a challenge due to the presence of ionizable amino acids in its chain. Such amino acids, such as arginine, cysteine, and tyrosine, require highly organic solvents to be solubilized and then allow the formation of nanofibers, however, many of these solvents are cytotoxic, compromising the manufacture of a biocompatible scaffold [64]. Currently, some acid solutions are being studied as less cytotoxic solvents for gelatin, in order to enable the development of gelatinous scaffolds through electrospinning. The produced matrix has advantages regarding the formation of a physical and biochemical environment suitable for cell growth and proliferation, however, it may compromise the mechanical properties of the tissue in vitro [24]. Fibrous gelatin scaffolds can have low mechanical properties, which can be a problem when considering muscle tissue engineering. This can be improved with a crosslinking process [64]. Gelatin can also be blended with synthetic polymers such as poly(aniline) (PANi), PCL, and PLGA to optimize their mechanical properties [24,65].

Fibrin is another natural polymer with potential use in tissue engineering. In addition to stimulating angiogenesis, this fibrillar protein is naturally involved in wound healing and tissue repair. Gilbert-Honick et al. [66] reported a fibrin scaffold, developed through electrospinning, promoted proliferation of human ADSCs on the fibers. The scaffold was designed to mimic the alignment and mechanical properties of native skeletal muscle. Although myogenesis was limited (cells with abundant actin and desmin cytoskeleton were found, but the formation of multinucleated cells was not observed), when implanted in a muscle defect model, the fibers integrated well with the native tissue, causing small scars, promoting long-term survival of transplanted ADSCs, and stimulating cellular and vascular ingrowth. Despite this, the amount of muscle fibers formed was small [66].

Other natural polymers, such as cellulose, chitin and its derivatives, and hyaluronic acid were studied as possible raw materials in the development of three-dimensional matrices and showed potential to induce cell differentiation and proliferation, despite their mechanical and morphological limitations. The polymers mentioned above are structural and quite common in the biological universe. Cellulose is the most abundant polymer in nature, and chitin is the second most abundant [67].

The good biological properties of cellulose have inspired the fabrication of scaffolds of the material blended with synthetic polymers such as polyurethane for muscle regeneration. The result was a matrix with improved mechanical properties. In this context, H9C2 cardiac myoblasts were cultured for 7 days. The elastomeric characteristics of cellulose/polyurethane were suitable for contractile tissues and myocytes showed an abundant actin cortex. The electrical behavior of the cells was not evaluated, but the myocytes expressed connexin-43, typical of gap junctions [68]. The results shown were promising, but further research needs to be conducted.

Chitin is commonly found in the exoskeleton of arthropods, it is commonly extracted from the shells of crustaceans and its main derivative, chitosan, is obtained by deacetylation of chitin. Chitosan is formed by natural cationic polysaccharides, being structurally similar to the glycosaminoglycans of the ECM [67]. Despite being biocompatible, their mechanical properties are lower than those of the body and thus hinders their use in tissue engineering applied to skeletal muscle, in which a higher modulus of elasticity is required. Recently, ADSCs cultured in chitosan arrays were able to regenerate rat urinary bladder smooth muscle [69]. However, the regeneration of skeletal muscle is still a challenge. On the other hand, hyaluronic acid is a polymer that already has clinical application in several situations, such as osteoarthritis, cartilage regeneration, anti-inflammatory effect, skin wound-healing, among others [70]. However, it is still being investigated for skeletal muscle regeneration. Although other studies have reported that hyaluronic acid has shown potential to be used as a scaffold, as it presents functional possibility to repair muscle in vivo, limitations in its process in electrospinning hinder the production of a matrix [24].

Hydrogels also present ideal characteristics for in vitro muscle tissue development. Formed by combinations of natural polymers, synthetic polymers, or both, hydrogels can be constructed to ensure specific conditions required for in vitro culture of certain cell types to repair muscle injuries [71]. Hydrogels present interesting advantages for use in cell delivery systems in muscle injuries or diseases, as they present similarities with the native tissue ECM, as well as less invasive approaches. In addition, hydrogels can be made to be completely degraded in the body, releasing growth factors that can stimulate the therapeutic process. Some gels are electrically conductive particularly for muscle tissue engineering, (e.g., gelatin methacryloyl and poly 3-thiophenacetic acid). The mechanical strength of hydrogels is low, but they can be associated with fibers produced by rotary-jet spinning or electrospinning, which brings an important gain in resistance [71].

Five natural polymeric materials in the form of hydrogels (collagen type I, agarose, alginate, fibrin, and collagen/chitosan) were evaluated as grafts for skeletal muscle tissue engineering. Both their mechanical properties and myogenic capacity were evaluated. Collagen, collagen/chitosan and fibrin showed high elasticity and formation of elongation structures, whereas agarose was the most brittle and alginate showed low ease of handling. Although collagen stimulated myogenesis, fibrin gels demonstrated greater myogenic potential, indicated by the expression of myogenin and in myosin heavy chain mRNA in satellite cells, along with the development of more extensive myotubes. Regarding the mechanical behavior of hydrogels with cells, collagen, collagen/chitosan, and fibrin showed high stretchability, agarose was the most brittle and alginate showed low strength [72].

Then, hydrogels composed of fibrin have shown good ability to promote activation, proliferation and differentiation of satellite cells in studies performed in mice. In order to increase the secretion of growth factors and proliferation of myoblasts, laminin was added to the fibrin hydrogel, and as a result, they obtained a tissue with more elongated and aligned cells after 24 h of culture. C2C12 myoblasts showed a significant increase in VEGF production and decrease in IL-6 production in laminin-enriched fibrin hydrogels compared to pure fibrin. An increase in the amount of MyoD and desmin, but a reduction in myogenin was also observed in myoblasts cultured in fibrin/laminin hydrogel. The combined application of electromechanical stimulation enhanced the production of VEGF and IGF-1 but not of differentiation markers [73].

Other studies have explored the potential of using fibrin hydrogels seeded with myoblasts directly into in situ defects in animal models. In these scaffolds exclusively formed by fibrin and myoblasts, there was less fibrosis and significant reconstitution of muscle tissue by the contribution of the transplanted cells in the regeneration and vascularization of the muscles. The results demonstrated myogenic cells can engraft and form new myofibers in the defect when casted along with fibrin gel [74].

Gelatin-based hydrogels have also been studied. Reports point to the development of a scaffold by using cross-linked gelatin to mimic the mechanical and biochemical characteristics of muscle, thus obtaining a matrix capable of inducing cell proliferation and differentiation in myotubes. It was demonstrated that this scaffold can be successfully implanted under the skin of mice, showing good biocompatibility and slow biodegradation rate. However, there was no mechanical or electrical stimulation of the hydrogels with cells, only their mechanical characterization prior to cell culture. Furthermore, the graft in the anterior tibial muscle did not impair muscle regeneration. Macrophages were also identified in the produced scaffold, which activate the inflammatory cascade and collaborate with the processes of differentiation of satellite cells into myofibers [75]. When developed through 3D bioprinting and inserted into the body, the hydrogels established direct cell-to-cell interaction, creating a cross-linked structure that ensures more efficient transport of nutrients and enzymes, thus enhancing the regenerative capacity of the in vitro tissue produced. This hydrogel model presents advantages regarding its “self-healing” mechanism; however, it requires the addition of substances, such as a porous or fibrous scaffold of polymeric material, to ensure an adequate structural and mechanical support to the tissue [24].

#### 3.2.3. Decellularized Extracellular Matrices (dECM)

As previously mentioned, another interesting approach to muscle reconstruction is the use of the natural structural scaffold of tissues, the ECM. Experimentally, a portion of tissue can be extracted, having its cells removed, but keeping the ECM intact, as far as possible, forming the so-called decellularized extracellular matrices (dECM) [76]. In general, dECMs can be obtained employing physicochemical agents, enzymes, detergents, or even a combination of these. These procedures may in fact alter the ECM, compromising its biochemical, mechanical, and structural hallmarks [53]. Despite that, these dECM can be used as a scaffold for tissue engineering or even be freeze-dried, ground, and used to form new hydrogels. The first reports of the use of dECM (obtained by segments of proximal jejunum from porcine prepared in saline and neomycin solution) in skeletal muscle were made by Clarke et al. [77], who showed good results in the reconstitution of abdominal wall muscles in dogs. It was shown a fast differentiation of cells with myogenic potential, besides low adherence, and a good incorporation of the dECM to the adjacent abdominal wall. The study also compared these results with a scaffold composed of a synthetic polymer, polypropylene. The results obtained with dECM were considered satisfactory, both for myogenesis and for immunological rejection [77].

In another approach, Perniconi et al. [78] presented the results of using a matrix obtained from adult sex-matched BALB/C mice muscle tissue, decellularized with 1% SDS in vitro for up to 48 h and implanted in rats. The results were very promising. Because it was a dECM from the skeletal muscle itself, the biological matrix, besides being able to promote myogenesis at the implant site, it also acted as an induction factor for angiogenesis around it. When transplanted in mice, the grafts were stable for several weeks, whilst being colonized by inflammatory and stem cells. Analyzing the data together, the scaffold produced from skeletal muscle dECM was shown to be a pro-myogenic environment, suggesting molecular properties of a niche for muscle differentiation by inducing host stem cells to myogenic specification.

On the other hand, other studies question the efficacy of dECM. In a very interesting comparative study, Aurora et al. [79] reported the use of MatriStem™ (a commercial dECM obtained from porcine urinary bladder) on muscle injuries in rats. It was shown that although vascularization and myogenesis were found to be restricted to the area near the injured site, no significant results were observed regarding remodeling of muscle shape and function (does not improve isometric force). In the authors’ view, the stimulation of skeletal muscle regeneration would occur through the production or migration of myogenic cells (such as satellite cells) to the site of injury. Therefore, it is implied that these dECM would be efficient only when there are still stem cells with myogenic determination present [79]. In theory, the dECM exists to provide a more physiological environment for the cells within. Perhaps, the choice of a dECM (in this case, urinary bladder) that is very different from the target tissue is a determining factor in the results presented by the respective authors. In addition, after tissue injury, the fibrotic process generated may give some support to the muscle structure, being beneficial to a specific extent. However, excessive fibrosis can bring harm to function [80].

When obtaining acellular scaffolds, the decellularization process of the matrices requires chemical and physical processes damaging the structural and biochemical composition necessary for the development of new muscle cells, thus compromising the regenerative capacity of tissues in vitro. Chen et al. [81], in a study with dECM obtained by pig bladders decellularized by 1.0% Triton X-100, noted that the presence of primary amines and carboxyls facilitate the adhesion of growth factors to the substrate. Nakayama et al. [82], in an interesting review, discus, even with a historical retrospective, factors such as mechanical and electrical stimulation in myogenesis, also highlighting the participation of several growth factors.

It was reported that VEGF, nerve growth factor (NGF), and glia-derived neurotrophic factor (GDNF) are essential for muscle tissue development, and thus should be preserved in the tissue produced. This scaffold conjugated with bFGF promoted human fibroblasts to proliferate in vitro and accelerated cellularization and vascularization after subcutaneous implantation. The biostability and mechanical strength of the scaffold could be increased with chemical crosslinking, which brought a gain in functional activity. However, factors such as rapid degradation and loss of bioactivity of the scaffold hinder the successful implantation of the in vitro developed tissue, so new physical and core-shell electrospun coating techniques are being studied in order to improve performance and preserve the desired characteristics of the scaffold [24]. In this context, materials produced with bioresorbable synthetic polymers appear.

In addition to tissue engineering methods that use cell cultures in matrices, more recent studies have described the recovery of rat muscle tissue by using an dECM obtained from skeletal muscle, without the need for cultured cells. In this model, implantation of the acellular scaffold at the injured site promoted both the migration of cells from the adjacent tissue into the matrix and the development and proliferation of these muscle cells in the implant region, enabling tissue regeneration [24].

The cellular response can vary in a relevant way, according to the characteristics of the scaffold. Cell surface glycans and glycosaminoglycans (GAGs) are cellular mediators, which regulate fundamental aspects of cellular survival. The glycocalyx, which surrounds all cells, actuates responses to growth factors, cytokines and morphogens at the cellular boundary, silencing or activating signaling pathways and gene expression. Thus, the interaction with the scaffold is quite specific [83]. This knowledge is important as it allows for the construction of scaffolds with more precise characteristics. In the case of skeletal muscle, MacQueen et al. [57] observed different cellular responses using gelatin fibrous matrices; with shorter fibers, large cell clumping was observed, whereas with long fibers, the cells showed much more aligned. In that same research, the authors used a smooth muscle cell line obtained from bovine aorta and a myoblast line from rabbit skeletal muscle. These cells were grown on gelatin fibers produced by rotary jet-spinning. Both cells showed good adhesion to the gelatin fibers, grew three-dimensionally from the matrix and formed focal points of adhesion. In the case of skeletal muscle cells, elongated cells with an abundant F-actin cytoskeleton and eccentric nuclei were observed, but mechanical testing demonstrated that cultured muscle lacked the mature contractile architecture observed in natural muscle [57].

#### 3.2.4. Bioresorbable Synthetic Polymers

Researchers study the development of scaffolds according to the physiological specificities of the region where they will be implanted to minimize the manipulation of natural polymers, as well as to obtain greater control of the processes that use synthetic polymers. Hydrogels produced with synthetic polymers, mainly bioresorbable, can be developed according to the particularities of the tissue to be regenerated. A very interesting report shows the production of a thick and well-vascularized muscle flap for reconstruction of full-thickness abdominal wall defects in rats [84]. Porous scaffolds made of poly (L-lactic acid) (PLLA) and PLGA in equal proportions were used. Myoblasts, human umbilical vein endothelial cells (HUVECs) and human dermal fibroblasts were cultured on this scaffold. The scaffold was maintained in vitro for cell expansion and then implanted around the femoral artery and vein. This allowed vessel growth before being transferred to reconstruct an abdominal wall defect. Extensive functional vascular density was observed in the scaffold. After transfer to the injured area, the muscle flaps proved to be well integrated with the surrounding tissue and had sufficient mechanical strength to support the abdominal viscera. Thus, the designed muscle flap proved to be an effective tool for reconstruction of large defects, avoiding the need for autologous flap removal and postoperative scarification [84].

This reinforces the possible therapeutic superiority of using a scaffold structure versus direct cell injection into the compromised area. In the article written by Boldrin et al. [85], lesions in the muscles of immuno-incompetent mice were generated and treated in two ways: (1) isolated myogenic cells (hMPCs); (2) PLGA bioresorbable scaffolds filled with the same myogenic cells. In form one, fibrous tissue was observed at the injury site. On the other hand, using the bioresorbable scaffold, the formation of muscle fibers was observed around injury. The differentiation markers used indicated myogenesis, but the cytoskeleton of the cells or their functional activity was not evaluated. The signs of muscle regeneration were noted only with the use of a scaffold associated with cells.

#### 3.2.5. Bioprinting

When the methods of 3D printing biological tissues were studied, the results presented the possibility of printing hydrogels capable of providing a matrix of differentiation and development of myoblasts such as the muscle matrix, which stimulates myogenesis in injured muscles. Kang et al. [54] also report that the use of the combination of natural and synthetic polymers, as the basis for making biopolymers, ensures both biocompatibility and satisfactory functional mechanical response in the regeneration of skeletal muscles. In general, the cells soaked in the bioinks (hydrogels with their own characteristics of viscosity and consistency) showed good viability and signs consistent with muscle differentiation. Some of the polymeric materials used for bioprinting were gelatin, methacryloyl, alginate, poly (ethylene glycol) (PEG), polyvinyl alcohol (PVA), often combined with each other and sometimes printed on dECM.

Since three-dimensional tissue bioprinting and electrospinning allow the creation of scaffolds with specific conditions in a controlled environment, such techniques have advantages when compared to dECM from biological donors, and present as promising methods for muscle regeneration when applied to tissue engineering [54].

From the information we have gathered in our study, muscles produced by tissue engineering are contractile and can be functional. Thus, an in vitro muscle fiber atrophy model was reported using the principles of tissue engineering. Primary mouse myoblasts were isolated from 6-week-old males and cultured on a scaffold formed of matrigel/type I collagen. In the structure formed, cells aligned on the substrate were observed, demonstrating contractile proteins. The muscle fragment produced was then shortened (from 25% to 50%) and atrophy was observed 6 days after the length reduction. Biochemical (number of total proteins and lactate production), histological, histochemical (for tropomyosin) and morphometric analyzes were then performed. Length reduction resulted in significant reductions in the average myofiber area (21.7%) and its content, total protein synthesis rate (22.0%), non-collagen protein content (6.9%), and force generation (50.4%). However, no significant change occurred in total metabolic activity or protein degradation rates [86]. However, the electrical behavior of muscle fibers was not evaluated.

In addition, it is worth noting that there are studies on muscle atrophy in the absence of gravitational force, i.e., targeting space flight experiments. Based on evidence that IGF-1 is shown to be an endogenous molecule that acts both in stimulating myoblast proliferation and myofiber hypertrophy and inhibiting the degradation of muscle mass proteins, Shansky et al. [87] studied the implantation of genetically engineered post-mitotic myofibers ex vivo as a method to deliver IGF-1 locally to skeletal muscle. The authors developed an in vitro tissue perfusion system manipulated into bioartificial muscles (BAMs) formed from silicone tubes filled with C2C12 myoblast and type I rat tail collagen mixture. That BAM system expressed recombinant human IGF-1 (rhIGF-1) and evaluated whether its release could stimulate muscle growth in non-genetically modified BAMs. The authors performed this modification in murine C2C12 myoblasts stably transduced with a retroviral vector to synthesize and secrete IGF-1. Positive results, that is, higher myofiber presence and larger myofiber cross-sectional area compared to the control group, were observed under the following conditions: from genetically modified C2 BAMs and implanted subcutaneously, as well as, under scaffold stress conditions. Interestingly, several cell concentrations were used in BAM. It was observed that, under mechanical stress, when starting with a greater cell quantity, there were a more accentuated loss of viability after 4 days of culture. Thus, under mechanical stress conditions, it was noted the ability of myoblasts to proliferate in the collagen gel to an optimal cell density before fusing into myofibers and maintain the functional structure.

The mechanical stimulation and electrical fields have also been tested to align the muscle cells in biomaterial scaffolds. These strategies could pre-align the muscle cells and improve their functionality in vitro; however, they only allowed micron-scale tissue or single-layered muscle bundle constructions that may be not suitable for treating extensive muscle defects [61]. Thus, the complexity of muscle tissue requires further research.

## 4. Conclusions

Over the years, research in the field of regenerative medicine has gained prominence and indicated promising results in the regeneration of skeletal striated muscle injuries. The studies reviewed in this paper explored different methods of development of scaffolds, polymers, and cell culture aiming to reconstruct in vitro muscle tissue and its environment in vivo. The different methodologies may account for the diverse results; however, certain combinations of materials are identified to be more interesting than others.

Methods developed by tissue engineering aiming to repair muscle injuries are relevant in seeking the improvement of quality of life and the greater autonomy of individuals with compromised motor capacity. In this aspect, techniques using bioresorbable materials, such as decellularized biological scaffolds and hydrogels, have shown promise in the regeneration of striated skeletal muscles, contributing to the recovery of motor capacity of individuals with physically disabled. This also includes relative emphasis on bio-printing, where the formulation of bioinks, besides allowing the printing of living cells, also offers biochemical and physical stimuli capable of proliferating and differentiating the target cells, and even guiding their migration. A bioprinted structure that mimics the muscle hierarchy and that can be responsive to mechanical or electrical stimuli appears to be a very promising path, however, we still don’t have such a structure yet.

The total regeneration of extensive muscle injuries is still a challenge. An old discussion that continues today converges on what would be the best form of implantation with cells in an in vivo situation: the inoculation of more differentiated cells with less proliferation power, but with greater functionality, or placing less differentiated cells, but that can multiply faster to fill the lesion and that differentiate at the lesion site. In the case of ADSCs, some data suggest that the use of more undifferentiated cells may have more interesting results than more differentiated cells.

## Figures and Tables

**Figure 2 bioengineering-09-00744-f002:**
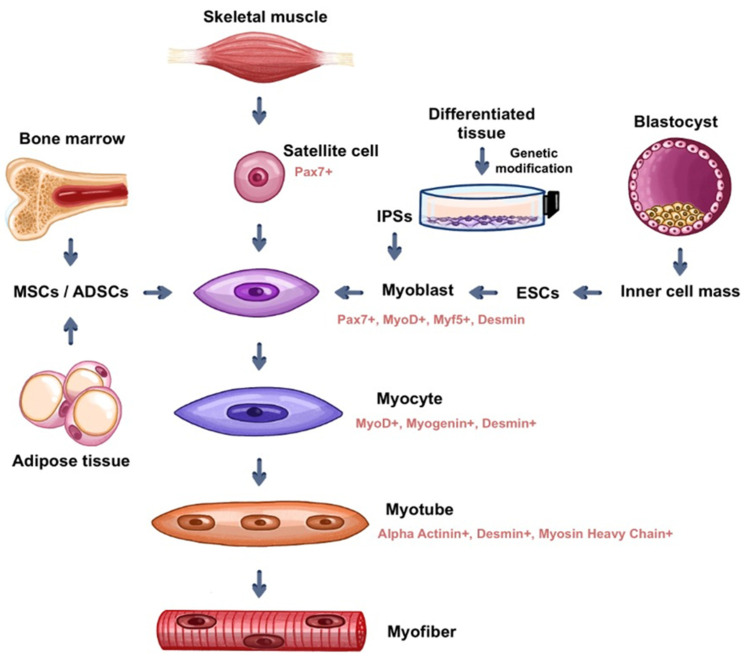
Myogenic cells most used in tissue engineering applied to skeletal muscle (source: prepared by the authors).

## Data Availability

The data described in this review are accessible from the corresponding author upon request.

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
