# Peer review of "Tissue Engineering Applied to Skeletal Muscle: Strategies and Perspectives"

_bioengineering, 2022, doi:10.3390/bioengineering9120744_

Round 1
Reviewer 1 Report
The review “Tissue engineering applied to skeletal muscle: strategies and perspectives” is very useful, combining the descriptions of the different cells employable and their characteristics, and the cell microenvironment requirements.
The manuscript however should be restructured and significantly enriched, structuring it in sub-paragraphs and inserting some more Tables and Figures.
In details:
line 148 – “bioactive factors”, are very important, should be described
3.1 Cells ….. sub-paragraphs for the different cells will be very useful
line 386 - dECM … please insert reference and shortly introduce the type: human and animal derived are both used (see for example Chem. Rev. 2020, 120, 19, 10608–10661)
3.3 The three-dimensional ….. also in this case sub-paragraphs for the different type
lines 447-452. Should be revised. The immune response occurs with natural biopolymers coming from different organisms, not with synthetic ones. Xenoantigens and microbial contamination is associated with natural biopolimers.
Instead, the concept that synthetic polymers lack of functional motifs recognized by the cells must be emphasized.
line 464 – synthesize biopolymers should be substituted with formulated biopolymers
line 466 – ….. polycaprolactone (PCL)
line 470 – with biochemical properties ……. I would suggest with structural properties
line 491 – 494 Please add further information, the 3 components are blended or crosslinked?
line 501 – acellular matrix …. I would suggest dECM for homogeneity
line 517 – require polar solvents ….. should be polar organic solvents? (Acetic acid, ethyl acetate …?)
line 524 - ….. associated….. should be blended or crosslinked, please verify
line 533- 539 there are some inconsistencies: cellulose has mechanical and morphological limitation in the first paragraph and with good mechanical properties in the second.
line 550 - …. hyaluronic acid has shown potential…. potential is not correct, it is already in clinic
line 596 – decellularized extracellular matrices (dECM)… substitute with dECM
line 629 – change in ……..chemical and physical processes damaging the structural and biochemical composition necessary …..
line 645 and subsequent – I suggest to insert observation on signaling molecules such as glycans inducing cell fate (see for example Cancer Science. 2021; 112:217–230)
line 660 – tissue flap ?
line 715 – bioartificial muscle … I would suggest explain in few lines its composition
Finally I suggest to add a list of acronyms and their meaning and verify in the text to use only the acronyms after the first mention.
Author Response
bioengineering-2001983
Referee responses
We thank the two referees who critically read our research. We try to answer all questions and accept all suggestions that have been made. Next, we will respond individually to each referee. Changes to the manuscript have been made in a different color font to be more easily located. We also inform you that the English of the text has been revised
Reviewer 1
The manuscript however should be restructured and significantly enriched, structuring it in sub-paragraphs and inserting some more Tables and Figures.
We thank the referee for evaluating our manuscript. Next, we will answer to the criticisms and suggestions made. Our responses and comments will be in blue font in the manuscript.
In details:
line 148 – “bioactive factors”, are very important, should be described
We thank the referee for the suggestion and accept it. The bioactive factors discussed in the cited study were added.
3.1 Cells ….. sub-paragraphs for the different cells will be very useful
We accept the referee's suggestion. We have separated the data on each cell population into a separate topic. We appreciate the suggestion
line 386 - dECM … please insert reference and shortly introduce the type: human and animal derived are both used (see for example Chem. Rev. 2020, 120, 19, 10608–10661)
The referee raised an important point. In fact, the results with dECM are very variable. We accepted the suggestion and, following the proposed model, we put information on the origin of the dECM and the general form of the method of obtaining it.
3.3 The three-dimensional ….. also in this case sub-paragraphs for the different type
We accept the referee's suggestion. The item is now subdivided for readability. We appreciate the comment.
lines 447-452. Should be revised. The immune response occurs with natural biopolymers coming from different organisms, not with synthetic ones. Xenoantigens and microbial contamination is associated with natural biopolimers. Instead, the concept that synthetic polymers lack of functional motifs recognized by the cells must be emphasized.
We apologize to the referee for our poor writing. We were actually referring to the inflammatory reaction, which is part of the innate immune response. We accept the suggestion. The sentence was rewritten and the indicated concept was incorporated into the text. We are grateful for this important highlight made by the reviewer.
line 464 – synthesize biopolymers should be substituted with formulated biopolymers
The referee is right. We accept and appreciate the suggestion.
line 466 – ….. polycaprolactone (PCL)
Polycaprolactone has already been abbreviated earlier on page 11. However, we now offer a list of abbreviations for easier reading.
line 470 – with biochemical properties ……. I would suggest with structural properties
We accept and appreciate the suggestion.
line 491 – 494 Please add further information, the 3 components are blended or crosslinked?
The correct word is blended. This has been corrected in the text. We thank the referee for the nomination.
line 501 – acellular matrix …. I would suggest dECM for homogeneity
We accept the suggestion. For standardization, "acellular matrix" has been replaced by dECM.
line 517 – require polar solvents ….. should be polar organic solvents? (Acetic acid, ethyl acetate …?)
To answer the referee's question, yes. We use this way of writing because it is the one that appears in the original article. But we accepted the suggestion and replaced it with organic solvents.
line 524 - ….. associated….. should be blended or crosslinked, please verify
The correct word is blended once again. It has been corrected. Thanks to the referee for the comment.
line 533- 539 there are some inconsistencies: cellulose has mechanical and morphological limitation in the first paragraph and with good mechanical properties in the second.
The referee is right. Our writing was not good. In fact, cellulose has interesting properties, but they can be improved. This is written in the first paragraph cited and is compatible with reference 67. What we try to say next is that cellulose blended with polyurethane forms a scaffold with better mechanical and biological properties. This was presented in the article cited in reference 68. I think this is clearer in the text. Thanks to the referee for the comment.
line 550 - …. hyaluronic acid has shown potential…. potential is not correct, it is already in clinic
The referee is right when he says that hyaluronic acid is a polymer that already has clinical application in several situations, such as osteoarthritis, cartilage regeneration, anti-inflammatory effect, skin wound-healing, among others. This is now in our text. What we originally tried to say is that for muscle tissue engineering it is still under study. We could not find a paper that showed the clinical use in this application. Thanks to the referee for the comment. Our wording has been modified to reflect the above information.
line 596 – decellularized extracellular matrices (dECM)… substitute with dECM
We accept the suggestion. We kept dECM written in full only the first time it appears in the text and as a sub-item of 3.3 (The three-dimensional environment for myogenesis). In the latter case, we understand that in the subtitle it is more elegant to write in full. We appreciate the suggestion.
line 629 – change in ……..chemical and physical processes damaging the structural and biochemical composition necessary …..
We accept the referee's wording suggestion. We appreciate the suggestion
line 645 and subsequent – I suggest to insert observation on signaling molecules such as glycans inducing cell fate (see for example Cancer Science. 2021; 112:217–230)
We appreciate the suggestion and accept it. The text now contains a general commentary on glycans and glycosaminoglycans as inducers of cell fate.
line 660 – tissue flap ?
Yes. In this case, we ask for the understanding of the referee. The authors of the original paper use this term even in the title of the article (Shandalov et al., 2014, reference 81). We are not comfortable with changing.
line 715 – bioartificial muscle … I would suggest explain in few lines its composition
We accepted the suggestion and briefly described the bioartificial muscle (BAM).
Finally I suggest to add a list of acronyms and their meaning and verify in the text to use only the acronyms after the first mention.
We accept the referee's suggestion. Provisionally (this list is not included in the Bioengineering template) we put it with an appendix, before the references. We understand that the journal staff will be able to help us with the right position.
Finally, we thank referee 1 for his criticisms and comments. Certainly, in answering it, our manuscript got better.

Reviewer 2 Report
I approached this review as someone with no prior knowledge of tissue engineering approaches to muscle, but as someone eager to learn. I must confess that I ended up a little unsatisfied. The paper provides what appears to be a comprehensive, if slightly uncritical, review of work on the isolation of various types of muscle and stem cells and their behaviour in synthetic and natural matrices. However, much of the information is rather generic and applies to most cell types and tissues. What I had hoped to learn was the specific challenges presented by muscle and the authors’ views on novel ways of addressing them.
I can perhaps best illustrate my concerns by raising some questions that occurred to me:
i. The introduction briefly alludes to the hierarchical structure of muscle. I believe, from my memory of early work attempting to transform skeletal muscle into cardiac muscle, that it is also quite plastic and can be modified by physiological factors. What levels of structure are sought in constructs and how does the “physiological” environment in which they are generated affect the outcomes?
ii. Presumably it is important that constructs develop appropriate passive and active mechanical properties. I am surprised that mechanics, either in characterising constructs or in the application of mechanical signals during growth is nowhere mentioned.
iii. The same comments apply to electrical signalling and stimulation.
iv. One of the essential features of the muscle cell is its active cytoskeleton, whose functioning depends, inter alia, on the correct interactions with the surrounding matrix. This is never even mentioned though I would have thought it may be a factor in selecting between the various scaffolds that have been used.
As it stands the manuscript is quite clear, though I would have welcomed more signposts, for example at the end of the introduction to section3, as to what is coming next – and why. The style also sometimes makes it difficult to understand what point is being made – for example, what are the punchlines of the discussion of methods of cell separation (P5) and the final paragraph of Sect 3.2?
Author Response
bioengineering-2001983
Referee responses
We thank the two referees who critically read our research. We try to answer all questions and accept all suggestions that have been made. Next, we will respond individually to each referee. Changes to the manuscript have been made in a different color font to be more easily located. We also inform you that the English of the text has been revised
Reviewer 2
Comments and Suggestions for Authors
I approached this review as someone with no prior knowledge of tissue engineering approaches to muscle, but as someone eager to learn. I must confess that I ended up a little unsatisfied. The paper provides what appears to be a comprehensive, if slightly uncritical, review of work on the isolation of various types of muscle and stem cells and their behaviour in synthetic and natural matrices. However, much of the information is rather generic and applies to most cell types and tissues. What I had hoped to learn was the specific challenges presented by muscle and the authors’ views on novel ways of addressing them.
We thank referee 2 for his criticisms. We sorry for hear that our manuscript did not meet the reviewer's expectations. Our text has been reworded a little, even to meet referee 1 as well. We tried to follow the path indicated by referee 2, increasing a little detail about signaling, mechanics and cytoskeleton. We hope that this new version will be more pleasant to the reviewer. All the answers we have given here and the text changes to suit referee 2 have been done in red font.
I can perhaps best illustrate my concerns by raising some questions that occurred to me:
We appreciate the suggestions and try to answer every point. But, the referee may be surprised, but not all articles address all the issues that were raised in the specific questions
- The introduction briefly alludes to the hierarchical structure of muscle. I believe, from my memory of early work attempting to transform skeletal muscle into cardiac muscle, that it is also quite plastic and can be modified by physiological factors. What levels of structure are sought in constructs and how does the “physiological” environment in which they are generated affect the outcomes?
Thanks to the referee for the comment. In fact, we tried to hierarchically show the structure of the muscle, making the transition to the cells present in the muscle and the environment around the muscle. Responding to the referee, in our humble opinion, yes, trying to reproduce the three-dimensional organization of muscle will have an important impact on tissue engineering results. In several examples in tissue engineering, attempts are made to mimic the tissue to be reconstructed. We understand that the same applies to muscle.
- Presumably it is important that constructs develop appropriate passive and active mechanical properties. I am surprised that mechanics, either in characterising constructs or in the application of mechanical signals during growth is nowhere mentioned.
However, we would like to make one caveat. Not all the articles we read deal with muscle mechanics.
iii. The same comments apply to electrical signalling and stimulation.
However, many articles we read deal with the issue of muscle signaling and stimulation.
- One of the essential features of the muscle cell is its active cytoskeleton, whose functioning depends, inter alia, on the correct interactions with the surrounding matrix. This is never even mentioned though I would have thought it may be a factor in selecting between the various scaffolds that have been used.
However, not all articles analyze in detail the cytoskeleton muscle cells in scaffolds.
As it stands the manuscript is quite clear, though I would have welcomed more signposts, for example at the end of the introduction to section3, as to what is coming next – and why. The style also sometimes makes it difficult to understand what point is being made – for example, what are the punchlines of the discussion of methods of cell separation (P5) and the final paragraph of Sect 3.2?
We understand the referee's point of view and accept the suggestion. We rewrote the end of the introduction to better contextualize our work and briefly present the points that will be discussed throughout the text. About the text on the page our objective was to say that, despite the advanced techniques, the absence of a specific myogenic marker decreases the efficiency of the process. The sentence has been rewritten to make this clearer. Thanks to the referee for the comment

Round 2
Reviewer 1 Report
now is suitable for publication
Reviewer 2 Report
The specific points I raised have been dealt with satisfactorily. I would have appreciated a more substantial to my more general concerns.